

# A new parametric information-gain criterion for tree-based machine learning algorithms

Diogo Costa[1], Vasco Vieira Costa[2] and Eugénio Rocha[1,2]

[1] Center for Research and Development in Mathematics and Applications (CIDMA), Universidade de Aveiro, Aveiro, Aveiro, Portugal
[2] Department of Mathematics, Universidade de Aveiro, Aveiro, Aveiro, Portugal

## ABSTRACT

Decision Trees (DTs) remain one of the most important algorithms in machine learning for their simplicity, interpretability, and often satisfactory performance. Furthermore, they are critical foundational components for more performant models such as Random Forests (RFs) and Gradient Boosted Trees. Central to DTs is the splitting process, where data is partitioned according to criteria traditionally based on information-theoretic measures such as Shannon entropy or Gini index. In this article, we propose a novel parametric entropy-based information gain criterion designed to generalize and extend classical entropic measures to improve classification performance in DTs and RFs. We introduce a five-parameter entropy formulation capable of replicating and extending known entropy measures. This new criterion was incorporated into DT and RF classifiers and evaluated on a collection of 18 benchmarking datasets, including both synthetic and real-world data retrieved from publicly available repositories. Performance was assessed using 5-fold cross-validation and optimized *via* Bayesian hyperparameter search, with weighted F1-score as the primary metric. Compared to splitting criteria based on existing entropy/purity measures (*e.g.*, Gini, Shannon, Rényi, and Tsallis), our method yielded statistically significant improvements in classification performance across most datasets. On multiclass and imbalanced datasets, such as the *Wine Quality* dataset, F1-score improvements exceeded 40% using RF algorithms. Bayesian signed-rank tests confirmed the robustness of our method, which never underperformed relative to standard approaches. The proposed entropy-based splitting criterion offers a flexible and effective alternative to classical information-theoretic measures, delivering improvements in classification performance.

# INTRODUCTION

Decision Trees (DTs) are still among the most widely used models in Machine Learning (ML), despite their roots being traced to as far back as the early 1960s (*Morgan & Sonquist, 1963*). This enduring relevance is owed to these models' underlying simplicity and practical performance. Succinctly, DTs operate by recursively splitting the dataset into subsets based on feature values, aiming to create groups that are as pure as possible

Corresponding author
Diogo Costa, d.costa@ua.pt

concerning a given target variable. Further aiding their adoption, the output of these recursive operations can be represented in a tree-like flowchart structure, greatly increasing their interpretability (*Kotsiantis, 2013*). Moreover, DTs also introduce a low computational cost and are compatible with both classification and regression tasks, which has justified their use in a wide range of applications, including, for example, in the medical, financial, and industrial sectors (*Costa & Pedreira, 2023*; *Mienye & Jere, 2024*).

Since their inception over half a century ago, DTs have lost their performance advantage compared to the best of the supervised learning approaches (*James et al., 2021*). However, they remain a critical foundational component of more powerful ensemble methods like Random Forests (RFs) and Gradient Boosted Trees, which are algorithms that remain highly competitive, particularly on tabular datasets, against even state-of-the-art neural network models (*Grinsztajn, Oyallon & Varoquaux, 2022*; *Uddin & Lu, 2024*). Moreover, there has been a newfound interest in the development of DTs due to their interpretability (*Hwang, Yeo & Hong, 2020*; *Hernández et al., 2021*), as the *black-box* nature commonly associated with most predictive and classifier models becomes an increasingly greater issue in an age where explainable machine learning is becoming a necessary condition (*Rudin, 2019*; *Roscher et al., 2020*).

Central to the induction (or construction) of DTs is the process of *recursive data partitioning* (*Costa & Pedreira, 2023*), which is guided by the output of a discrete function over the input attributes (*Kotsiantis, 2013*). The selection of the most appropriate function is typically determined by some splitting criteria, such as information gain, the Gini value, or the Gain ratio (GR), with the selection of these criteria directly influencing the structure and performance of the resulting tree. In the more (now) classical implementations of DT models, such as the Iterative Dichotomiser 3 (ID3) algorithm (*Quinlan, 1986*), information-theoretic measures (*e.g.*, Shannon entropy) have well-established roles as splitting criteria. Nonetheless, past research, such as the work done by *Nowozin (2012)*, shows that potential limitations remain in standard methods, as their performance is not uniform across different domains, data distributions, or learning objectives.

Recently, there has been additional research in either the development of novel splitting criteria (*Leroux, Boussard & Dès, 2018*; *Ayllón-Gavilán et al., 2025*; *Hwang, Yeo & Hong, 2020*; *Loyola-González, Ramírez-Sáyago & Medina-Pérez, 2023*) or in re-adapting existing methods with less common information-theoretic measures (*Ignatenko, Surkov & Koltcov, 2024*; *Nowozin, 2012*; *Maszczyk & Duch, 2008*; *Gajowniczek, Zabkowski & Orłowski, 2015*). In these works, the goal becomes the development of alternative and corrected measures aimed at improving split balance, reducing overfitting, and enhancing the generalization of the output DTs, with the adequate selection of the splitting criteria becoming particularly important in the context of high-dimensional, imbalanced, or noisy datasets. In a detailed analysis provided in *Hernández et al. (2021)*, it is shown that no one split evaluation measure is capable of consistently outperforming all others. Alternatively, one approach is to combine multiple evaluation measures and select the candidate splits that better adapt to the input data (*Loyola-González, Ramírez-Sáyago & Medina-Pérez, 2023*).

However, in this article, we introduce an alternative perspective, based on the development of a more generic notion of entropy that is more parametric than existing

approaches, and that enables the generalization of existing information-theoretic criteria most commonly used as the basis of splitting procedures. Through the added degrees of freedom provided by this approach, it is possible to construct splits that better adjust to input data. This method, evaluated for both open data repositories and synthetic datasets, is seen to achieve performance improvements of over 40% in F1-score relative to standard methods in common datasets such as the 'Wine Quality' dataset, when using RF algorithms. Moreover, we provide a Bayesian statistical analysis to assess that the proposed method presents a statistically significant improvement while never underperforming, with at worst, matching the performance of the best standard model.

The remainder of this article is organized as follows. 'Decision Trees for Classification', provides a generic discussion regarding DTs, their construction, and their splitting criteria. 'Generalized Entropy', introduces the 5-parameter expression for the generalization of entropy and its formulation into a target function for its use as a DT splitting criterion. 'Description of Computational Experiments', describes the datasets used for computational experiments, experimental methodology, and result evaluation metrics. 'Results of Computational Experiments', provides the summarized results obtained from the computational experiments and discusses the implementation of a Bayesian statistical analysis to validate the quality of the introduced method. 'Discussion', interprets the obtained results and statistical analysis. 'Conclusions', provides the concluding remarks regarding the introduced method. 'Proofs of Known Entropies', shows the mathematical proof of the generalizations possible to achieve with the presented parametric entropy expression. 'Complete Results for Proposed Method', provides additional tabular results.

# DECISION TREES FOR CLASSIFICATION

Multiple versions of the DT algorithm exist; however, most notable are perhaps their most "traditional" forms as seen in the case of the original ID3 (*Quinlan, 1986*), its successor C4.5 (*Quinlan, 1993*), and the Classification and Regression Tree (CART) (*Breiman et al., 1984*). In these forms, the structure of a DT consists of internal and terminal nodes. The former represent logical tests (*splits*) where each split has a binary outcome (true or false); while the latter are the *leaves* of the tree and correspond to output predictions (*Costa & Pedreira, 2023*). DTs in formulations such as CART are compatible with classification and regression tasks, hence leaves can correspond to either labels or constant numbers. These values are selected during the induction of the tree, where, starting at a root node that contains the full training dataset, this input space is recursively partitioned into homogeneous regions with respect to a given target variable. This forms the basis for the *splitting* process.

In the case of ID3, C4.5, and CART, they all take on greedy approaches to constructing DTs (*Han, Kamber & Pei, 2012*), meaning that they will evaluate all possible splits across all features and select the one that best separates the data according to a predefined criterion. The most commonly used criteria include information gain (based on information-theoretic measures), the Gini index, and the gain ratio. These measures assess the *purity* of the resulting subsets, aiming to maximize class purity in the child nodes, where a pure partition would mean that every element would have the same label.

Information gain (IG), used in ID3, is calculated as the reduction in entropy, *e.g.*, using the Shannon entropy, before and after the split. Mathematically, this corresponds to computing (*Mienye & Jere, 2024*)

$$IG(S, A) = H(S) - \sum_{v \in Values(A)} \frac{|S_v|}{|S|} H(S_v), \tag{1}$$

where $S$ is our dataset, $A$ is an attribute of $S$, $Values(A)$ are the unique values in attribute $A$, $S_v$ the subset of $S$ for which attribute $A$ takes on value $v$, and $H$ the entropy function, which in the case of Shannon's entropy is given by *Shannon (1948)*

$$H(S) = -\sum_{i=1}^{n} P_i \log_2 P_i, \tag{2}$$

where $n$ is the number of unique classes in the set $S$ and $P_i$ is the proportion of the samples in the set that belong to class $i$. Essentially, the higher the information gain, the more adequate an attribute is in partitioning the data, as the subsets become more homogeneous. In more advanced DT algorithms, such as C4.5, *Gain Ratio* is used, a method where IG is normalized in order to correct its bias toward attributes with many distinct values (*Quinlan, 1993*; *Han, Kamber & Pei, 2012*). The gain ratio for a dataset $S$ and an attribute $A$ is given by

$$GainRatio(A) = \frac{IG(A)}{SplitInfo(S, A)}, \tag{3}$$

where

$$SplitInfo(S, A) = -\sum_{v \in Values(A)} \frac{|S_v|}{|S|} \log_2 \left( \frac{|S_v|}{|S|} \right). \tag{4}$$

Lastly, the final of the most common splitting criteria is the Gini index (or value or impurity), employed by the CART algorithm. This metric measures the probability of misclassifying a randomly chosen instance from the dataset and is defined as *Jost (2006)* and *Kotsiantis (2013)*

$$Gini(S) = 1 - \sum_{i=1}^{n} P_i^2. \tag{5}$$

Once the best split is identified, the dataset is divided accordingly, and the algorithm recurses on each subset. This process continues until a stopping criterion is met, such as reaching a maximum tree depth, achieving a minimum number of samples per node, or obtaining pure nodes (*Nowozin, 2012*). In classification tasks, each leaf node is assigned the majority class of the samples it contains.

Currently, DTs are seldom used in isolation, being much more commonly employed as a component of ensemble models. The most well-known of these is perhaps Random Forests (RFs), a method based on the induction of multiple DTs and then combining their outputs to improve predictive accuracy and control overfitting (*Breiman, 2001*). This

algorithm operates by training each tree on a different bootstrap sample (random sample with replacement) of the dataset and using a random subset of features at each split to ensure diversity among trees. For classification, predictions are made by majority voting across trees; for regression, predictions are averaged. This combination of bagging (bootstrap aggregating) and random feature selection makes RFs more robust to noise, resistant to overfitting, and effective for high-dimensional data. A more comprehensive insight into these methods can be found in works such as *Mienye & Jere (2024)* and *Costa & Pedreira (2023)*.

The focus of this article lies in the development of novel criteria for the splitting of tree-based algorithms and on surpassing some of the limitations that remain within the standard methods. In this sense, there have been previous attempts at this issue. For instance, in *Leroux, Boussard & Dès (2018)*, a balanced gain ratio is discussed, aimed at addressing the bias towards unbalanced splits in the GR method used by *Quinlan (1986)*, essentially by correcting the split information by a constant value. Empirically evaluated, this approach is seen to limit the depth of resulting trees with an improvement in classification accuracy. In other settings, splitting criteria are adapted to serve particular tasks. For instance, in *Ayllón-Gavilán et al. (2025)*, a splitting criterion is defined for use with ordinal classification. Another example can be found in *Hwang, Yeo & Hong (2020)* where the main goal is not to achieve the best performance, but rather to lead to the creation of the most interpretable tree possible.

More in line with the motivation of this article is the work done by *Nowozin (2012)*, where the classical entropy estimators, such as Shannon entropy or Gini index, are replaced, in this case, with the Grassberger entropy estimator, enabling an increase in predictive performance during classification tasks. We can further see the use of nonclassical entropies for the induction of DTs in *Gajowniczek, Zabkowski & Orłowski (2015)* and *Maszczyk & Duch (2008)*. In both of these past examples, the authors employed Rényi and Tsallis parametric entropies. Introduced by *Rényi (1961)*, the Rényi entropy of order $\alpha$ of a set $S$, and with $0 < \alpha < \infty$ and $\alpha \neq 1$, is defined as

$$H_\alpha(S) = \frac{1}{1-\alpha} \log\left(\sum_{i=1}^{n} P_i^\alpha\right). \tag{6}$$

In case $\alpha \in \{0, 1, \infty\}$, it is defined as

$$H_\alpha(S) = \lim_{x \to \alpha} H_x(S). \tag{7}$$

Rényi's entropy generalizes various other notions of entropy. For instance, as $\alpha \to 1$ (notation meaning: when $\alpha$ tends to 1 by a valid branch), the Shannon entropy is recovered. Alternatively, the Tsallis entropy of a set $S$, introduced by *Tsallis (1988)*, is defined as

$$S_q(S) = k \frac{1}{q-1}\left(1 - \sum_{i=1}^{n} P_i^q\right), \tag{8}$$

where $k \in \mathbb{R}^+$ and $q \in \mathbb{R}$. Tsallis' entropy is also capable of recovering other entropic definitions. For example, as $q \to 1$, the Boltzmann–Gibbs entropy is obtained.

Advancements are still occurring in this field, namely in the use of dynamically adjustable criteria at split time (*Loyola-González, Ramírez-Sáyago & Medina-Pérez, 2023*), or in regards to the use of deformed entropies for the improvement of target functions. One such recent example of the latter can be found in the work by *Ignatenko, Surkov & Koltcov (2024)*, where the potential use of nonclassical entropies for the computation of information gain on RFs under classification and regression tasks was studied. In this work, the nonstandard entropies enabled substantial performance gains (in terms of accuracy) of models, for the application of Rényi, Tsallis, and Sharma–Mittal entropies. The Sharma–Mittal entropy (*Akturk, Bagci & Sever, 2007*) is defined as

$$S_{SM}(S) = \frac{1}{1-r}\left[\left(\sum_{i=1}^{n} P_i^q\right)^{\frac{1-r}{1-q}} - 1\right], \tag{9}$$

and is further capable of retrieving the Rényi entropy for $r \to 1$ and the Tsallis entropy for $q \to r$.

Although these previous works are comparable in motivation to our work, in the sense that the goal is improving the splitting process of tree-based algorithms through the use of nonstandard entropies, the implementation substantially differs. Here, the approach is to develop a new parametric expression that encompasses the classical methods (*e.g.*, retrieving Shannon entropy or Gini index); however, by the nature of additional degrees of freedom, it enables the deduction of additional criteria that may be more suitable for separating data at each split.

## GENERALIZED ENTROPY

This section will now introduce the 5-parameter expression for the generalization of entropy, first introduced in the authors' past work in the scope of data complexity estimation (*Costa, Rocha & Ferreira, 2024*). Consider a set of probabilities $P = (P_1, \ldots, P_n) \in [0,1]^n$ with $\sum_{i=1}^{n} P_i = 1$, the generalized entropy $\hat{E}$ is defined by

$$\hat{E}_{\zeta_1,\zeta_2}^{\alpha,\beta,\gamma}(P) = \left(\hat{\varphi}_{\zeta_1,\zeta_2}^{\alpha,\beta,\gamma}(P) - k_0\right)(k_1 - k_0)^{-1}, \tag{10}$$

where $(\alpha, \beta, \gamma, \zeta_1, \zeta_2) \in (\mathbb{R}_0^+)^5$ are adequate parameters,

$$\hat{\varphi}_{\zeta_1,\zeta_2}^{\alpha,\beta,\gamma}(P) = \zeta_2 - \ln\left(\sum_{i=1}^{n} P_i^\alpha\left[-\zeta_1 - \ln\left(P_i^\beta\right)\right]^\gamma\right), \tag{11}$$

and the $q$-logarithm, for any $q \in \mathbb{R}_0^+ \setminus \{1\}$ and $x > 0$, is given by

$$q-\ln(\mathrm{x}) = \frac{x^{1-q} - 1}{1-q}. \tag{12}$$

For $q = 1$, the $q-\ln(x)$ coincides with the natural logarithm $\ln(x)$, by computing the limit as $q \to +1$, using the L'Hôpital's rule, and the fact that $d\, x^{1-q}/dq = -\ln(x)\,x^{1-q}$. Constants $k_0$ and $k_1$ represent the minimum and maximum theoretical values of $\hat{\varphi}$,

**Table 1 Required parameter configurations to obtain the most common generalizations of impurity/entropy.**

| Parameter | | | | | Generalization |
|---|---|---|---|---|---|
| $\alpha$ | $\beta$ | $\gamma$ | $\zeta_1$ | $\zeta_2$ | |
| 0 | 2 | 1 | 0 | 0 | Gini impurity |
| 1 | 1 | 1 | 1 | 0 | Shannon entropy |
| $w$ | 0 | 0 | 0 | 1 | Rényi entropy $R_w$ |
| $w$ | 0 | 0 | 0 | 0 | Tsallis entropy $T_w$ |
| $q$ | 0 | 0 | 0 | $\frac{r-q}{1-q}$ | Sharma–Mittal entropy |

respectively. Accordingly, $k_0 = min(\hat{\varphi})$, obtained whenever $\exists_j : P_j = 1$. Alternatively, $k_1 = max(\hat{\varphi})$, occurring whenever each event is equally probable, *i.e.*, $P_i = \frac{1}{n}$. Table 1 showcases the required parameter re-configuration to achieve a normalized version of the most common generalizations of impurity and entropy used as criteria for splitting in DTs. Proofs for the retrieval of these entropic measures using $\hat{E}$ are given in the Supplemental File 'Proofs of Known Entropies'. Note, however, that the list shown in Table 1 is not comprehensive, in the sense that there are further entropic definitions that can still be retrieved. Besides generalizing to other entropic estimators, $\hat{E}$ is capable of further obtaining other entropy estimations through parametric variation. In Fig. 1, some of these behaviors are plotted.

To employ the measure $\hat{E}$ as a splitting criterion for DTs in classification tasks, it is first necessary to formulate a target function. Considering (1), the target function will be formulated in a similar way as seen in the work by *Ignatenko, Surkov & Koltcov (2024)*. In the case $S_j$ is the set of data points falling into node $j$, then $IG_j = IG\left(S_j, S_j^L, S_j^R\right)$, where $S_j^L$ and $S_j^R$ are the subsets which fall into the left and right subtrees, respectively. Therefore, information gain is given by

$$IG_j = \hat{E}(S_j) - \sum_{i \in \{L,R\}} \frac{|S_j^i|}{|S_j|} \hat{E}\left(S_j^i\right). \tag{13}$$

As such, when employing this definition of information gain, each of the five parameters accepted by $\hat{E}$ will become a new hyperparameter of the DT. To help guide hyperparameter tuning, a sensitivity analysis was conducted, focusing on parameters $\alpha$, $\beta$, and $\gamma$. Parameters $\zeta_1$ and $\zeta_2$ were defined as $\zeta_1, \zeta_2 \in \{0, 1\}$, as this would simplify the analysis whilst maintaining the generic properties of $\hat{E}$ intact. The analysis was two-stage. First, the Morris screening method (*Morris, 1991*; *Campolongo, Cariboni & Saltelli, 2007*) was applied to estimate the mean absolute effect, $\mu*$, of each parameter in overall importance, and the standard deviation, $\sigma$, to quantify nonlinearity and interaction effects. Secondly, a Sobol variance-based sensitivity analysis (*Sobol, 2001*; *Saltelli, 2002*; *Saltelli et al., 2010*) was performed to compute both first-order (S1) indices to measure main effects and total-effect (ST) indices to capture the combined impact of main and interaction effects. Both of these methods were applied using the implementation made available in the SALib library

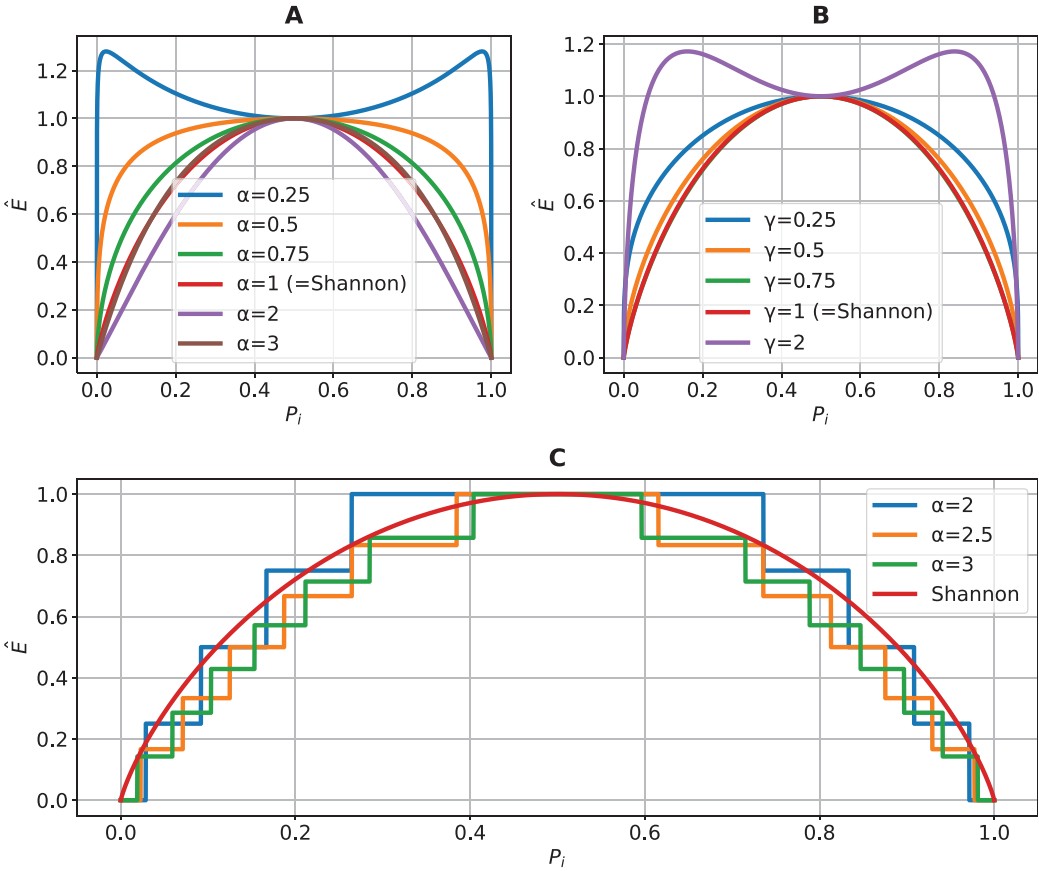

**Figure 1 Illustration of the behavior of the proposed generalized entropy function Ê for a random variable with two possible outcomes against $P_i$, where $= [P_i, 1 - P_i]$, showing how the entropy value varies with different parameter settings.** In (A), parameters $\{\gamma = 1; \beta = 1; \zeta_1 = 1; \zeta_2 = 0\}$ are fixed, and $\alpha$ is varied; this includes the Shannon entropy as the special case $\alpha = 1$. In (B), parameters $\{\alpha = 1; \beta = 1; \zeta_1 = 1; \zeta_2 = 0\}$ are fixed, and $\gamma$ is varied; again recovering Shannon entropy at $\gamma = 1$. In (C), parameters $\{\gamma = 1; \beta = 0; \zeta_1 = 1; \zeta_2 = 0\}$ are fixed, and $\alpha$ is varied with $\alpha > 1$; the Shannon entropy is plotted alongside for reference. These visualizations demonstrate the behavior of the proposed entropy function with changing data distributions and its ability to recover classical entropy measures as special cases.

(*Herman & Usher, 2017*; *Iwanaga, Usher & Herman, 2022*) (SALib version 1.4.7 (https://github.com/salib/salib)).

The comparison of the results yielded from both methods (shown in Figs. 2 and 3) enables the identification of the parameters with strong and stable influences *vs.* those whose effects are driven primarily by interactions. In this case, no single parameter exerts a predominant influence through main effects, as indicated by the low first-order indexes. While both $\alpha$ and $\gamma$ exhibit the highest first-order influences, these values remain considerably less than the total-effect indexes. Suggesting that the behavior of $\hat{E}$ is mostly driven by interaction between parameters. The results of the Morris analysis for $\beta$ appear to suggest a high degree of influence; however, this assumption is contradicted by the Sobol method, thus suggesting that its interactions are less pervasive or more localized.

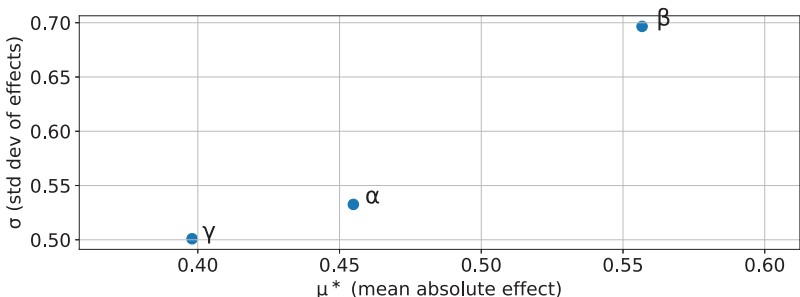

**Figure 2 Morris screening results showing mean absolute effect $\mu^*$ vs. standard deviation $\sigma$ for parameters $\alpha$, $\beta$, and $\gamma$.** $\mu^*$ indicates overall parameter influence, while $\sigma$ reflects nonlinearity and interaction strength. Parameters in the top-right are both influential and involved in interactions.

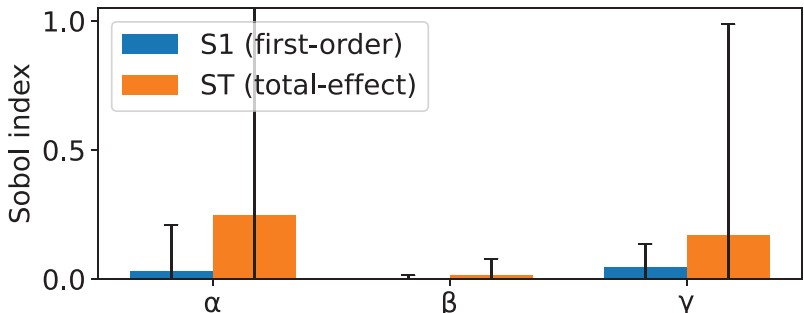

**Figure 3 Sobol sensitivity indices for parameters $\alpha$, $\beta$, and $\gamma$, showing first-order effects (S1) and total effects (ST).** S1 measures the proportion of output variance explained by each parameter alone, while ST captures both main and interaction effects. Large gaps between ST and S1 indicate strong interactions.

## DESCRIPTION OF COMPUTATIONAL EXPERIMENTS

The proposed splitting criterion was evaluated for classification tasks across two distinct groups of datasets, namely datasets retrieved from open repositories and synthetically generated datasets. In the case of the former, these were retrieved from both the UCI (https://archive.ics.uci.edu) and OpenML (https://www.openml.org) repositories, and their characteristics are shown in Table 2. In the case of the latter, they were generated using the `make_classification` function found in scikit-learn (*Pedregosa et al., 2011*) (scikit-learn version 1.6.1 (https://scikit-learn.org/1.6/api/index.html)), and their description is shown in Table 3. Dataset selection took into account the following requirements: (i) be reasonably broad in terms of application/problem areas; (ii) have a reasonably comprehensive combination between number of features/number of instances across datasets; (iii) validate for both binary and multiclass datasets; (iv) validate for both balanced and imbalanced datasets (in the case of imbalanced datasets, with varying degrees of imbalance); and, (v) contain both numerical (continuous), ordinal, and categorical features across datasets.

Two algorithms were implemented: (i) a (classification) DT, based on the conventional ID3 architecture, which supported both numeric and categorical features and used as its

**Table 2 Description of datasets sourced from open repositories.** The shown ID is relative to their respective repositories.

| Repository | Name | ID | # of features | # of instances | # of classes | Feature type | Class proportions (# of instances/class) |
|---|---|---|---|---|---|---|---|
| UCI | Breast Cancer Wisconsin | 15 | 9 | 699 | 2 | Ordinal | 458:241 |
| | Iris | 53 | 4 | 150 | 3 | Continuous | 50:50:50 |
| | Spambase | 94 | 57 | 4,601 | 2 | Continuous | 2,788:1,813 |
| | Statlog (Shuttle) | 148 | 7 | 58,000 | 7 | Continuous | 45,586:8,903:3,267:171:50:13:10 |
| | Wine Quality | 186 | 11 | 4,898 | 11 | Continuous | 2,836:2,138:1,079:216:193:30:5 |
| | Students' Dropout and Academic Success | 697 | 36 | 4,424 | 3 | Continuous, Categorical, Ordinal | 2,209:1,421:794 |
| OpenML | mfeat-morphological | 18 | 7 | 2,000 | 10 | Continuous, Categorical | 200:200:200:200:200:200:200:200:200:200 |
| | diabetes | 37 | 9 | 768 | 2 | Continuous, Ordinal | 500:268 |
| | wdbc | 1510 | 31 | 569 | 2 | Continuous | 357:212 |
| | wilt | 1570 | 6 | 4,839 | 2 | Continuous | 4,578:261 |
| | Titanic | 40704 | 4 | 2,201 | 2 | Categorical, Ordinal | 1,490:711 |
| | dataset_31_credit-g | 42633 | 21 | 1,000 | 2 | Categorical | 700:300 |
| | phoneme | 44087 | 6 | 3,172 | 2 | Continuous | 1,586:1,586 |

**Table 3 Description of synthetically generated datasets.** All of the synthetic datasets are class-balanced and contain only continuous features.

| Name | # of features | # of informative features | # of instances | # of classes |
|---|---|---|---|---|
| synth_1 | 10 | 2 | 100 | 2 |
| synth_2 | 10 | 5 | 100 | 4 |
| synth_3 | 5 | 2 | 1,000 | 2 |
| synth_4 | 10 | 5 | 1,000 | 2 |
| synth_5 | 10 | 5 | 1,000 | 4 |

target function the equation defined in Eq. (13); and (ii) a RF based on an ensemble of DTs with feature bagging and voting based on the most common label values. The source code for the algorithms used in these experiments was made available (https://doi.org/10.5281/zenodo.15241909), and was developed with standardization in mind, following the conventions of the popular Python package of scikit-learn, making it compatible with its methods. The fixed parameters for DTs were: 2 samples as the minimum number required to split an internal node; and, minimum impurity decrease set as 0. The foundational DTs for the RFs were constructed with these same fixed parameters.

In terms of non-fixed parameters, for the tests focused on the DTs, their maximum depth was varied as $max\_depth \in \{10, 50, 100, 200\}$. In the case of RFs, the number of estimators in each forest was varied as $n\_estimators \in \{10, 25, 50\}$, and the maximum depth of individual trees was varied as $max\_depth \in \{10, 25\}$. The search space for the parameters in $\hat{E}$ was constructed considering preliminary empirical findings, and was defined as: $\alpha \in \{0, 1, 2, 3\}$; $\beta \in [0, \ldots, 3]$, with a step size of 0.1; $\gamma \in [0, \ldots, 3]$, with a step size of 0.1; $\zeta_1 \in \{0, 1, 2, 3\}$; and, $\zeta_2 \in \{0, 1\}$. These search spaces were selected based on the prior sensitivity analysis, where the effect of parameter $\alpha$ was comparatively stable

while $\beta$ and $\gamma$ produce large variability in effects predominantly through interactions. Consequently, a more coarse grid was used for $\alpha$ to concentrate computational effort where the proposed entropic function $\hat{E}$ is most interaction-sensitive, reducing the risk of missing narrow high-performance regions while keeping total evaluations manageable.

The search for the best-performing hyperparameters for the proposed splitting criterion (*i.e.*, for parameters $\alpha, \beta, \gamma, \zeta_1, \zeta_2$) was conducted for all combinations of maximum depth and number of estimators in the DTs and RFs algorithms, where applicable. Due to the elevated number of test combinations and datasets, a Bayesian search was conducted over the search space, using the `BayesSearchCV` class in scikit-optimize (https://scikit-optimize.github.io) (scikit-optimize version 0.10.2), with a set number of 100 iterations (*i.e.*, 100 parameters sampled per dataset per maximum depth, and, in the case of RFs, per number of estimators). Furthermore, these parameters were evaluated through 5-fold cross-validation with the target metric set to weighted F1-score, given by

$$F1_{\text{weighted}} = \sum_{i=1}^{N} \left( \frac{n_i}{\sum_{j=1}^{N} n_j} \times F1_i \right), \tag{14}$$

where $N$ is the number of classes, $n_i$ represents the support for the $i$-th class, and $F1_i$ is the F1-score for the $i$-th class, computed through

$$F1 = \frac{2\,\text{TP}}{2\,\text{TP} + \text{FP} + \text{FN}}, \tag{15}$$

where TP is the number of true positives, FP is the number of false positives, and FN is the number of false negatives. In addition to the weighted F1-score used for model selection during the hyperparameter search, complementary evaluation metrics were computed, namely Balanced Accuracy, Precision, and Recall. Balanced Accuracy is defined as

$$\text{Balanced Accuracy} = \frac{1}{N} \sum_{i=1}^{N} \frac{\text{TP}_i}{\text{TP}_i + \text{FN}_i}, \tag{16}$$

where $\text{TP}_i$ and $\text{FN}_i$ are the true positives and false negatives for the $i$-th class, respectively. The Precision and Recall for each class are computed as:

$$\text{Precision} = \frac{\text{TP}}{\text{TP} + \text{FP}}, \tag{17}$$

$$\text{Recall} = \frac{\text{TP}}{\text{TP} + \text{FN}}. \tag{18}$$

These metrics were also averaged in a weighted manner according to class support, ensuring that the performance evaluation reflects the influence of class distribution in the dataset.

The usage of the weighted F1-score is uncommon in comparable works found in the literature, which tend to favor reporting accuracy measurements (see, for instance, the experimental methodology in *Nowozin (2012)* or *Ignatenko, Surkov & Koltcov (2024)*). However, this is disingenuous as the datasets selected for testing frequently show some degree of class imbalance. For example, the 'Statlog (Shuttle)' dataset, used in testing in

multiple comparable works (including the two previously mentioned), is a seven-class set with the majority class representing nearly 80% of instances. Hence, the weighted F1-score is selected as the performance evaluation metric for this work, and thus any mention of F1-score will be regarding its weighted implementation. The weighted version was selected (instead of macro or micro extensions) as a compromise, somewhat attenuating the effect of very low support classes negatively skewing results, while still providing enough sensitivity for penalizing misclassifications in imbalanced datasets.

The introduced parametric entropies were benchmarked against the known splitting criteria given by the Gini index, Shannon entropy, Rényi entropy, and Tsallis entropy, as these are the most utilized in comparable works. The benchmarks were obtained using the same tree structures and parameters as the introduced method. In the case of Rényi, parameter $\alpha$ in Eq. (6) was set as $\alpha = 2$, and, in the case of Tsallis, parameters $k$ and $q$ in Eq. (8) were set as $k = 1$, and $q = 2$.

## RESULTS OF COMPUTATIONAL EXPERIMENTS

To aid in interpreting results, in this section only a summary presentation is given (shown in Table 4), where, for each result obtained, two additional metrics were computed: *Lowest Improvement* (LI), and *Highest Improvement* (HI). Respectively, these correspond to the lowest and highest percentual gain in F1-scoring of the parametric entropies obtained using $\hat{E}$ compared to the known entropic measures. The complete results for the computational experiments can be seen in the Supplemental File 'Complete Results for Proposed Method', these include: results for DTs (shown in Tables S1, S2, S3 and S4); results for RFs using a maximum depth of 10 for the DTs (shown in Tables S5, S6, and S7); and, results for RFs using a maximum depth of 25 for the DTs (shown in Tables S8, S9, and S10).

To more effectively assess the method's performance, the parameter combination yielding the highest weighted F1-score for each dataset in the hyperparameter search was selected. This optimal configuration was subsequently compared with the known entropy values. The evaluation was carried out using a class-wise stratified 5-fold cross-validation procedure. In this approach, the dataset is split into five equally sized folds while preserving the proportion of classes in each fold. For each iteration, the model is trained on four folds and tested on the remaining one. This process is repeated five times so that each fold serves once as the test set, and the results are then averaged to obtain the final performance metrics. The results are presented in terms of the *Highest Improvement* and *Lowest Improvement* observed. For the DT algorithms, these results are shown in Tables 5, 6, and 7, which correspond to Balanced Accuracy, Precision, and Recall, respectively. The equivalent results for the RF algorithms are reported in Tables 8, 9, and 10.

### Bayesian statistical analysis

Comparing the performance of two cross-validated classifiers is traditionally achieved using Student's t-test (*Nadeau & Bengio, 2003*) for the case of comparisons over a single dataset, or using the Wilcoxon signed-ranked test (*Demsar, 2006*; *Nowozin, 2012*) in the case of comparisons over multiple datasets. Both these tests are based on the

**Table 4 Summarized results obtained for both the DT and RF algorithms.** Shown are the percentage of *Lowest Improvement* (LI) and of *Highest Improvement* (HI) of the mean results for the weighted F1-score over the 5 folds (for each dataset). In this sense, LI is the worst-case scenario improvement, comparing the proposed method $\hat{E}$, with the best performing of the classical measures. Comparably, HI represents the best-case scenario improvement (compared with the lowest-performing classical measure).

| Dataset | Decision Tree | | | | | | | | Random Forest | | | | | | | | | | | |
| --- | --- | --- | --- | --- | --- | --- | --- | --- | --- | --- | --- | --- | --- | --- | --- | --- | --- | --- | --- | --- |
| Depth estimators | 10 | | 50 | | 100 | | 200 | | 10 | | | | | | 25 | | | | | |
| | | | | | | | | | 10 | | 25 | | 50 | | 10 | | 25 | | 50 | |
| | LI (%) | HI (%) | LI (%) | HI (%) | LI (%) | HI (%) | LI (%) | HI (%) | LI (%) | HI (%) | LI (%) | HI (%) | LI (%) | HI (%) | LI (%) | HI (%) | LI (%) | HI (%) | LI (%) | HI (%) |
| BreastCancer | 1.463 | 1.985 | 1.463 | 1.985 | 1.463 | 1.985 | 1.463 | 1.985 | 1.030 | 1.648 | 0.617 | 1.233 | 0.206 | 1.336 | 0.414 | 1.344 | 0.617 | 1.028 | 0.515 | 1.133 |
| Iris | 0 | 2.976 | 0 | 2.976 | 0 | 2.976 | 0 | 2.976 | 0 | 3.594 | 1.458 | 2.917 | 1.469 | 3.673 | 1.469 | 4.302 | 0 | 3.673 | 0.735 | 2.099 |
| Spambase | 0.325 | 0.650 | 1.816 | 3.312 | 0.863 | 2.373 | 1.289 | 2.793 | 1.732 | 3.580 | 0.449 | 1.798 | 0.114 | 1.362 | 1.606 | 4.358 | 1.575 | 2.925 | 0.454 | 2.497 |
| Statlog | 1.807 | 4.570 | 0 | 0.502 | 0 | 0.401 | 0 | 0.401 | 5.079 | 12.817 | 10.508 | 14.640 | 9.211 | 11.842 | 9.032 | 15.778 | 5.216 | 12.320 | 10.560 | 15.194 |
| Wine Quality | 3.473 | 10.603 | 5.960 | 13.411 | 2.689 | 11.092 | 3.477 | 12.252 | 13.688 | 37.262 | 18.333 | 36.667 | 18.116 | 38.768 | 20.398 | 33.167 | 22.742 | 40.323 | 23.344 | 40.536 |
| Students | 1.910 | 3.138 | 3.626 | 5.718 | 3.905 | 5.718 | 3.905 | 5.718 | 2.806 | 5.613 | 2.541 | 3.587 | 0.607 | 2.580 | 3.254 | 4.882 | 1.783 | 2.972 | 0.448 | 2.990 |
| mfeat-morphological | 1.429 | 4.143 | 3.730 | 6.743 | 3.736 | 7.615 | 1.903 | 5.857 | 3.698 | 18.182 | 2.115 | 7.855 | 4.405 | 7.636 | 3.762 | 7.837 | 1.997 | 7.373 | 3.598 | 8.546 |
| diabetes | 4.110 | 6.164 | 5.772 | 8.322 | 3.306 | 5.923 | 5.391 | 7.951 | 1.955 | 6.917 | 2.392 | 5.381 | 0.459 | 6.279 | 5.365 | 10.283 | 3.907 | 7.236 | 2.534 | 5.067 |
| wdbc | 0.528 | 2.218 | 0.842 | 2.526 | 0.946 | 2.629 | 0.946 | 2.629 | 0 | 1.572 | 0.415 | 1.350 | 1.037 | 1.971 | 0.210 | 2.306 | 0 | 0.935 | 0.208 | 1.249 |
| wilt | 0.204 | 0.714 | 0.204 | 0.613 | 0.204 | 0.613 | 0.204 | 0.613 | 0.217 | 0.217 | 0 | 0 | 0 | 0 | 0 | 0.108 | 0 | 0.108 | 0 | 0.108 |
| Titanic | 0 | 0 | 0 | 0 | 0 | 0 | 0 | 0 | 4.836 | 11.807 | 5.299 | 10.190 | 2.286 | 2.714 | 4.911 | 9.686 | 4.959 | 6.336 | 2.680 | 3.385 |
| dataset_31_credit-g | 2.374 | 4.050 | 2.917 | 4.306 | 3.719 | 5.096 | 3.851 | 5.227 | 1.495 | 9.716 | 1.468 | 2.936 | 1.000 | 2.667 | 3.324 | 10.983 | 1.286 | 4.341 | 2.114 | 2.764 |
| phoneme | 0 | 1.425 | 0.476 | 2.143 | 0.476 | 2.143 | 1.065 | 2.722 | 1.225 | 3.554 | 0.845 | 1.932 | 1.202 | 2.644 | 2.589 | 4.192 | 1.086 | 2.654 | 0.601 | 1.803 |
| synth_1 | 4.040 | 4.040 | 4.040 | 4.040 | 4.040 | 4.040 | 4.040 | 4.040 | 3.061 | 8.673 | 3.000 | 6.000 | 3.000 | 5.000 | 4.040 | 7.172 | 2.020 | 3.030 | 1.000 | 5.000 |
| synth_2 | 18.584 | 27.655 | 22.851 | 31.027 | 22.851 | 31.027 | 22.851 | 31.027 | 0 | 19.770 | 1.342 | 24.385 | 14.675 | 36.688 | 12.646 | 27.635 | 5.139 | 27.409 | 0 | 26.389 |
| synth_3 | 0.209 | 1.885 | 0 | 1.157 | 0 | 1.157 | 0 | 1.157 | 0.735 | 2.731 | 0.520 | 1.873 | 0.208 | 0.833 | 0.950 | 1.795 | 0.730 | 1.564 | 0.418 | 1.461 |
| synth_4 | 0 | 1.896 | 0.589 | 2.473 | 0.236 | 2.128 | 0.354 | 2.243 | 3.496 | 5.868 | 1.695 | 4.722 | 0.957 | 3.469 | 2.270 | 5.422 | 2.881 | 3.962 | 1.437 | 2.156 |
| synth_5 | 2.358 | 5.548 | 2.107 | 6.320 | 1.135 | 5.390 | 1.554 | 5.791 | 4.249 | 8.801 | 4.403 | 7.812 | 4.172 | 9.179 | 5.113 | 15.038 | 3.026 | 7.349 | 3.329 | 4.993 |

**Table 5 Summarized results of balanced accuracy obtained for the DT algorithm, when using the best found parameter configuration after tuning for the weighted F1-score.** Shown are the percentages of LI and HI of the mean results over the 5 folds (for each dataset).

| Dataset | Depth | | | | | | | |
|---|---|---|---|---|---|---|---|---|
| | **10** | | **50** | | **100** | | **200** | |
| | LI (%) | HI (%) | LI (%) | HI (%) | LI (%) | HI (%) | LI (%) | HI (%) |
| BreastCancer | 1.998 | 2.679 | 1.998 | 2.679 | 1.998 | 2.679 | 1.998 | 2.679 |
| Iris | 2.174 | 2.920 | 2.174 | 2.920 | 2.174 | 2.920 | 2.174 | 2.920 |
| Spambase | 0.243 | 0.691 | 1.822 | 3.540 | 0.819 | 2.521 | 1.165 | 2.873 |
| Statlog | 5.711 | 6.411 | 1.019 | 12.035 | 2.431 | 9.123 | 2.580 | 9.282 |
| Wine Quality | 7.744 | 16.516 | 15.554 | 35.840 | 7.285 | 26.680 | 7.483 | 28.863 |
| Students | 2.254 | 3.208 | 3.458 | 6.545 | 3.338 | 5.984 | 3.362 | 6.009 |
| mfeat-morphological | 2.632 | 2.632 | 2.291 | 5.695 | 3.372 | 7.779 | 1.722 | 6.058 |
| diabetes | 2.614 | 5.425 | 6.495 | 10.759 | 4.256 | 8.430 | 5.777 | 10.013 |
| wdbc | 0.758 | 2.443 | 0.862 | 2.548 | 1.180 | 2.872 | 1.180 | 2.872 |
| wilt | 1.420 | 2.893 | 0.978 | 2.445 | 0.978 | 2.445 | 0.978 | 2.445 |
| Titanic | 0 | 0 | 0 | 0 | 0 | 0 | 0 | 0 |
| dataset_31_credit-g | 0.148 | 3.199 | 1.303 | 2.447 | 2.817 | 5.798 | 1.464 | 4.405 |
| phoneme | 1.135 | 1.483 | 0.490 | 2.147 | 0.527 | 2.185 | 1.057 | 2.724 |
| synth_1 | 4.211 | 4.211 | 4.211 | 4.211 | 4.211 | 4.211 | 4.211 | 4.211 |
| synth_2 | 18.421 | 32.353 | 28.947 | 44.118 | 28.947 | 44.118 | 28.947 | 44.118 |
| synth_3 | 0.207 | 1.904 | 0.643 | 1.179 | 0.631 | 1.166 | 0.631 | 1.166 |
| synth_4 | 0.004 | 1.933 | 0.577 | 2.548 | 0.210 | 2.173 | 0.339 | 2.305 |
| synth_5 | 2.225 | 5.701 | 1.970 | 6.834 | 0.981 | 5.798 | 1.398 | 6.235 |

**Table 6 Summarized results of precision obtained for the DT algorithm, when using the best found parameter configuration after tuning for the weighted F1-score.** Shown are the percentages of LI and HI of the mean results over the 5 folds (for each dataset).

| Dataset | Depth | | | | | | | |
|---|---|---|---|---|---|---|---|---|
| | **10** | | **50** | | **100** | | **200** | |
| | LI (%) | HI (%) | LI (%) | HI (%) | LI (%) | HI (%) | LI (%) | HI (%) |
| BreastCancer | 1.375 | 2.049 | 1.375 | 2.049 | 1.375 | 2.049 | 1.375 | 2.049 |
| Iris | 1.391 | 1.942 | 1.391 | 1.942 | 1.391 | 1.942 | 1.391 | 1.942 |
| Spambase | 0.3 | 0.621 | 1.819 | 3.382 | 0.862 | 2.411 | 1.209 | 2.763 |
| Statlog | 0.814 | 2.514 | 0.049 | 0.475 | 0.009 | 0.415 | 0.018 | 0.423 |
| Wine Quality | 0.843 | 8.628 | 6.403 | 13.469 | 2.648 | 11.222 | 3.487 | 12.595 |
| Students | 2.782 | 4.081 | 4.652 | 7.005 | 3.180 | 5.256 | 3.204 | 5.280 |
| mfeat-morphological | 2.716 | 7.648 | 5.382 | 9.250 | 8.365 | 13.303 | 1.574 | 6.203 |
| diabetes | 5.193 | 7.714 | 6.165 | 9.408 | 3.545 | 6.708 | 5.625 | 8.852 |
| wdbc | 0.484 | 2.367 | 0.798 | 2.686 | 0.744 | 2.632 | 0.744 | 2.632 |
| wilt | 0.243 | 0.678 | 0.251 | 0.649 | 0.251 | 0.649 | 0.251 | 0.649 |
| Titanic | 0 | 0 | 0 | 0 | 0 | 0 | 0 | 0 |

| Dataset | Depth | | | | | | | |
|---|---|---|---|---|---|---|---|---|
| | **10** | | **50** | | **100** | | **200** | |
| | LI (%) | HI (%) | LI (%) | HI (%) | LI (%) | HI (%) | LI (%) | HI (%) |
| dataset_31_credit-g | 1.784 | 3.960 | 1.967 | 3.948 | 3.512 | 5.523 | 3.196 | 5.2 |
| phoneme | 1.227 | 1.481 | 0.472 | 2.099 | 0.529 | 2.156 | 1.024 | 2.659 |
| synth_1 | 3.645 | 3.645 | 3.645 | 3.645 | 3.645 | 3.645 | 3.645 | 3.645 |
| synth_2 | 25.937 | 38.572 | 27.612 | 43.238 | 27.612 | 43.238 | 27.612 | 43.238 |
| synth_3 | 0.173 | 1.818 | 0.609 | 1.123 | 0.568 | 1.081 | 0.568 | 1.081 |
| synth_4 | 1.936 | 1.936 | 0.383 | 2.580 | 0.030 | 2.220 | 0.118 | 2.309 |
| synth_5 | 2.716 | 5.796 | 2.585 | 7.061 | 1.657 | 6.093 | 2.019 | 6.471 |

**Table 7 Summarized results of recall obtained for the DT algorithm, when using the best found parameter configuration after tuning for the weighted F1-score.** Shown are the percentages of LI and HI of the mean results over the 5 folds (for each dataset).

| Dataset | Depth | | | | | | | |
|---|---|---|---|---|---|---|---|---|
| | **10** | | **50** | | **100** | | **200** | |
| | LI (%) | HI (%) | LI (%) | HI (%) | LI (%) | HI (%) | LI (%) | HI (%) |
| BreastCancer | 1.516 | 1.978 | 1.516 | 1.978 | 1.516 | 1.978 | 1.516 | 1.978 |
| Iris | 2.174 | 2.920 | 2.174 | 2.920 | 2.174 | 2.920 | 2.174 | 2.920 |
| Spambase | 0.283 | 0.640 | 1.820 | 3.409 | 0.875 | 2.449 | 1.229 | 2.809 |
| Statlog | 1.421 | 3.508 | 0.048 | 0.457 | 0.002 | 0.394 | 0.010 | 0.399 |
| Wine Quality | 1.756 | 7.892 | 5.318 | 11.580 | 2.054 | 9.186 | 3.411 | 10.983 |
| Students | 2.013 | 3.944 | 4.222 | 6.772 | 5.255 | 7.443 | 5.288 | 7.476 |
| mfeat-morphological | 2.632 | 2.632 | 2.291 | 5.695 | 3.372 | 7.779 | 1.722 | 6.058 |
| diabetes | 5.572 | 7.763 | 6.496 | 8.919 | 3.352 | 5.703 | 6.120 | 8.534 |
| wdbc | 0.567 | 2.276 | 0.941 | 2.657 | 0.943 | 2.659 | 0.943 | 2.659 |
| wilt | 0.211 | 0.701 | 0.211 | 0.658 | 0.211 | 0.658 | 0.211 | 0.658 |
| Titanic | 0 | 0 | 0 | 0 | 0 | 0 | 0 | 0 |
| dataset_31_credit-g | 3.863 | 5.217 | 5.180 | 6.250 | 5.036 | 6.105 | 5.755 | 6.831 |
| phoneme | 1.135 | 1.482 | 0.491 | 2.147 | 0.528 | 2.185 | 1.056 | 2.722 |
| synth_1 | 4.211 | 4.211 | 4.211 | 4.211 | 4.211 | 4.211 | 4.211 | 4.211 |
| synth_2 | 18.421 | 32.353 | 28.947 | 44.118 | 28.947 | 44.118 | 28.947 | 44.118 |
| synth_3 | 0.210 | 1.921 | 0.635 | 1.170 | 0.635 | 1.170 | 0.635 | 1.170 |
| synth_4 | 1.932 | 1.932 | 0.592 | 2.536 | 0.237 | 2.174 | 0.355 | 2.295 |
| synth_5 | 2.263 | 5.702 | 2.0 | 6.886 | 1.0 | 5.838 | 1.429 | 6.287 |

Null-Hypothesis Significance Test (NHST), which has been argued in *Benavoli et al. (2017)* to be inadequate in the comparison of classifiers, particularly when their performance is near equivalent (possibility of draws). As such, to avoid these shortcomings, here we will take inspiration from the analysis conducted in *Hernández et al. (2021)* through the application of Bayesian statistical analysis methods for the comparison of ML models, as

**Table 8 Summarized results of balanced accuracy obtained for the RF algorithm, when using the best found parameter configuration after tuning for the weighted F1-score.** Shown are the percentages of LI and HI of the mean results over the 5 folds (for each dataset).

| Dataset | 10 | | | | | | 25 | | | | | |
|---|---|---|---|---|---|---|---|---|---|---|---|---|
| Depth estimators | 10 | | 25 | | 50 | | 10 | | 25 | | 50 | |
| | LI (%) | HI (%) | LI (%) | HI (%) | LI (%) | HI (%) | LI (%) | HI (%) | LI (%) | HI (%) | LI (%) | HI (%) |
| BreastCancer | 0.224 | 1.667 | 0.007 | 0.649 | 0.119 | 0.119 | 1.180 | 1.180 | 0.423 | 1.290 | 0.526 | 0.526 |
| Iris | 1.460 | 1.460 | 0.704 | 2.878 | 0.714 | 1.439 | 0.719 | 4.478 | 0 | 0 | 0.714 | 1.439 |
| Spambase | 1.146 | 1.146 | 1.214 | 1.214 | 1.432 | 2.410 | 1.257 | 1.257 | 0.567 | 2.841 | 0.285 | 2.514 |
| Statlog | 15.051 | 56.367 | 0.791 | 12.131 | 9.526 | 22.152 | 7.614 | 16.073 | 10.970 | 35.527 | 9.785 | 26.675 |
| Wine Quality | 16.573 | 29.998 | 15.264 | 35.021 | 19.801 | 46.307 | 51.036 | 88.412 | 55.639 | 102.701 | 54.338 | 97.229 |
| Students | 0 | 0 | 0.566 | 1.297 | 0.580 | 0.580 | 0.193 | 3.998 | 0.955 | 2.245 | 0.390 | 1.684 |
| mfeat-morphological | 0.976 | 4.370 | 3.343 | 9.237 | 2.161 | 7.953 | 3.292 | 7.173 | 1.615 | 2.483 | 0.601 | 4.528 |
| diabetes | 5.041 | 10.8 | 0.010 | 2.650 | 1.566 | 1.566 | 1.769 | 6.213 | 0.707 | 3.686 | 1.331 | 1.941 |
| wdbc | 0.562 | 0.878 | 0.646 | 0.854 | 0.262 | 0.766 | 0 | 0 | 0.152 | 0.556 | 0.309 | 0.390 |
| wilt | 0 | 0 | 0 | 0 | 0 | 0 | 0.380 | 0.380 | 0.376 | 0.376 | 0.377 | 0.377 |
| Titanic | 7.110 | 7.110 | 0.587 | 0.587 | 0.055 | 0.361 | 1.847 | 2.370 | 0.418 | 8.352 | 0.121 | 0.636 |
| dataset_31_credit-g | 2.289 | 3.151 | 1.276 | 1.276 | 0.419 | 1.458 | 0.474 | 3.872 | 0 | 0 | 0.563 | 0.563 |
| phoneme | 0.239 | 1.670 | 0.038 | 0.273 | 0.192 | 0.891 | 0.555 | 3.465 | 0 | 0 | 0.114 | 0.114 |
| synth_1 | 1.064 | 7.955 | 2.128 | 2.128 | 2.105 | 2.105 | 2.151 | 7.955 | 0 | 0 | 1.042 | 2.105 |
| synth_2 | 0 | 0 | 0 | 0 | 0 | 0 | 8.108 | 8.108 | 6.452 | 6.452 | 2.564 | 11.111 |
| synth_3 | 0.5 | 2.385 | 0.113 | 0.316 | 0 | 0 | 0.430 | 0.430 | 0.519 | 1.919 | 0.094 | 0.423 |
| synth_4 | 1.281 | 2.120 | 2.299 | 2.299 | 2.059 | 3.559 | 1.577 | 3.615 | 0.364 | 2.388 | 0 | 0 |
| synth_5 | 0.817 | 4.559 | 0 | 0 | 2.486 | 6.199 | 10.052 | 12.678 | 0.202 | 0.202 | 1.441 | 2.167 |

**Table 9 Summarized results of precision obtained for the RF algorithm, when using the best found parameter configuration after tuning for the weighted F1-score.** Shown are the percentages of LI and HI of the mean results over the 5 folds (for each dataset).

| Dataset | 10 | | | | | | 25 | | | | | |
|---|---|---|---|---|---|---|---|---|---|---|---|---|
| Depth estimators | 10 | | 25 | | 50 | | 10 | | 25 | | 50 | |
| | LI (%) | HI (%) | LI (%) | HI (%) | LI (%) | HI (%) | LI (%) | HI (%) | LI (%) | HI (%) | LI (%) | HI (%) |
| BreastCancer | 0.280 | 1.352 | 0.104 | 0.537 | 0.006 | 0.173 | 0.615 | 0.615 | 0.269 | 0.866 | 0.227 | 0.227 |
| Iris | 1.256 | 1.256 | 0.637 | 1.827 | 0.356 | 1.528 | 0.856 | 3.286 | 0 | 0 | 0.082 | 0.929 |
| Spambase | 0.572 | 0.572 | 0.435 | 0.435 | 0.794 | 1.335 | 0.799 | 0.799 | 0.191 | 1.308 | 0.029 | 1.208 |
| Statlog | 2.733 | 13.548 | 4.281 | 7.797 | 1.482 | 20.6 | 1.034 | 2.501 | 4.670 | 6.982 | 2.885 | 5.179 |
| Wine Quality | 0.474 | 27.1 | 11.910 | 39.5 | 19.167 | 45.395 | 4.645 | 51.624 | 9.330 | 35.216 | 11.167 | 50.746 |
| Students | 0 | 0 | 0.136 | 6.086 | 0.659 | 0.892 | 1.870 | 5.320 | 0.409 | 1.408 | 0.570 | 4.405 |
| mfeat-morphological | 0.818 | 7.128 | 4.433 | 8.806 | 4.668 | 6.939 | 2.228 | 6.798 | 1.234 | 3.546 | 0.134 | 0.350 |
| diabetes | 6.410 | 11.507 | 2.217 | 2.217 | 2.687 | 3.296 | 2.365 | 7.2 | 2.280 | 4.052 | 1.813 | 5.232 |
| wdbc | 0.435 | 0.524 | 0.5 | 0.528 | 0.197 | 0.495 | 0 | 0 | 0.138 | 0.450 | 0.135 | 0.381 |
| wilt | 0 | 0 | 0 | 0 | 0 | 0 | 0.020 | 1.193 | 1.177 | 1.177 | 0.020 | 0.020 |
| Titanic | 1.845 | 25.020 | 1.559 | 1.559 | 0.341 | 0.641 | 1.246 | 14.221 | 3.699 | 4.256 | 0.3 | 0.3 |
| dataset_31_credit-g | 3.248 | 3.850 | 6.862 | 6.862 | 1.532 | 4.795 | 1.061 | 7.128 | 1.203 | 1.203 | 0.031 | 10.180 |
| phoneme | 1.190 | 1.718 | 0.007 | 0.007 | 0.013 | 0.582 | 0.538 | 3.295 | 0 | 0 | 0.093 | 0.093 |

| Dataset | 10 | | | | | | 25 | | | | | |
|---|---|---|---|---|---|---|---|---|---|---|---|---|
| Depth estimators | 10 | | 25 | | 50 | | 10 | | 25 | | 50 | |
| | LI (%) | HI (%) | LI (%) | HI (%) | LI (%) | HI (%) | LI (%) | HI (%) | LI (%) | HI (%) | LI (%) | HI (%) |
| synth_1 | 0.350 | 8.270 | 1.757 | 2.280 | 1.743 | 1.937 | 1.972 | 7.004 | 0 | 0 | 0.785 | 1.743 |
| synth_2 | 7.965 | 7.965 | 1.764 | 1.764 | 3.135 | 3.135 | 0.640 | 15.250 | 19.820 | 19.820 | 7.574 | 28.616 |
| synth_3 | 0.365 | 2.097 | 0.112 | 0.307 | 0 | 0 | 0.231 | 0.231 | 0.458 | 1.849 | 0.105 | 0.518 |
| synth_4 | 0.054 | 1.333 | 2.407 | 2.407 | 2.023 | 3.210 | 1.325 | 3.320 | 0.452 | 2.254 | 0 | 0 |
| synth_5 | 1.426 | 6.108 | 0 | 0 | 2.794 | 6.984 | 8.851 | 13.087 | 0.224 | 0.224 | 0.398 | 2.143 |

**Table 10 Summarized results of recall obtained for the RF algorithm, when using the best found parameter configuration after tuning for the weighted F1-score.** Shown are the percentages of LI and HI of the mean results over the 5 folds (for each dataset).

| Dataset | 10 | | | | | | 25 | | | | | |
|---|---|---|---|---|---|---|---|---|---|---|---|---|
| Depth estimators | 10 | | 25 | | 50 | | 10 | | 25 | | 50 | |
| | LI (%) | HI (%) | LI (%) | HI (%) | LI (%) | HI (%) | LI (%) | HI (%) | LI (%) | HI (%) | LI (%) | HI (%) |
| BreastCancer | 0.298 | 1.359 | 0.150 | 0.596 | 0.002 | 0.152 | 0.001 | 0.753 | 0.295 | 0.894 | 0.148 | 0.148 |
| Iris | 1.460 | 1.460 | 0.704 | 2.878 | 0.714 | 1.439 | 0.719 | 4.478 | 0 | 0 | 0.714 | 1.439 |
| Spambase | 0.926 | 0.926 | 0.824 | 0.824 | 1.089 | 1.845 | 0.987 | 0.987 | 0.397 | 2.039 | 0.173 | 1.853 |
| Statlog | 3.860 | 8.295 | 0.021 | 3.307 | 2.843 | 4.760 | 2.873 | 4.864 | 5.782 | 9.301 | 4.722 | 8.347 |
| Wine Quality | 7.561 | 12.798 | 6.827 | 19.283 | 9.536 | 25.987 | 13.059 | 27.313 | 16.629 | 35.921 | 17.573 | 36.318 |
| Students | 0 | 0 | 0.640 | 0.640 | 0.320 | 0.320 | 1.671 | 4.303 | 0.282 | 1.590 | 0.346 | 0.536 |
| mfeat-morphological | 0.976 | 4.370 | 3.343 | 9.237 | 2.161 | 7.953 | 3.292 | 7.173 | 1.615 | 2.483 | 0.601 | 4.528 |
| diabetes | 3.984 | 6.816 | 1.715 | 1.715 | 0.570 | 1.337 | 1.307 | 4.041 | 1.285 | 2.818 | 0.565 | 2.281 |
| wdbc | 0.564 | 0.565 | 0.549 | 0.553 | 0.185 | 0.554 | 0 | 0 | 0.187 | 0.553 | 0.002 | 0.370 |
| wilt | 0 | 0 | 0 | 0 | 0 | 0 | 0.022 | 0.022 | 0.022 | 0.022 | 0.022 | 0.022 |
| Titanic | 4.093 | 4.093 | 0.927 | 0.927 | 0.061 | 0.061 | 1.119 | 2.129 | 0.980 | 3.451 | 0.122 | 0.306 |
| dataset_31_credit-g | 1.288 | 1.724 | 0.714 | 0.714 | 0.142 | 0.712 | 0.422 | 3.030 | 0 | 0 | 0.142 | 0.571 |
| phoneme | 0.237 | 1.669 | 0.040 | 0.273 | 0.192 | 0.889 | 0.555 | 3.466 | 0 | 0 | 0.116 | 0.116 |
| synth_1 | 1.064 | 7.955 | 2.128 | 2.128 | 2.105 | 2.105 | 2.151 | 7.955 | 0 | 0 | 1.042 | 2.105 |
| synth_2 | 0 | 0 | 0 | 0 | 0 | 0 | 8.108 | 8.108 | 6.452 | 6.452 | 2.564 | 11.111 |
| synth_3 | 0.536 | 2.404 | 0.105 | 0.316 | 0 | 0 | 0.431 | 0.431 | 0.526 | 1.919 | 0.105 | 0.423 |
| synth_4 | 1.339 | 2.159 | 2.302 | 2.302 | 2.063 | 3.571 | 1.579 | 3.624 | 0.372 | 2.402 | 0 | 0 |
| synth_5 | 0.810 | 4.538 | 0 | 0 | 2.5 | 6.187 | 10.017 | 12.650 | 0.148 | 0.148 | 1.439 | 2.174 |

proposed in *Benavoli et al. (2017)* (which is a recommended work for additional details). Namely, we will employ the Bayesian signed-rank test, which enables the comparison of cross-validation performance of two classifiers for a group of datasets, effectively replacing the traditionally used Wilcoxon signed-rank test.

Succinctly, the Bayesian signed-rank test functions by comparing two classifier models ($M_1$ and $M_2$) for a selected performance measure (in this case, weighted F1-score) over multiple cross-validation folds and multiple datasets. The output for the Bayesian test is a

**Table 11 Bayesian signed-rank test for the DT algorithms across cross-validation results of all datasets for the weighted F1-score.** Here, ROPE = {0.01, 0.02}; ε = 0.95; **W** represents a significant win for $\hat{E}$; **D** represents a significant draw for $\hat{E}$; and, **L** represents a significant loss for $\hat{E}$. For the case of significant wins, the posterior probability value is shown in bold.

| C | Depth | ROPE = 1% | | | | ROPE = 2% | | | |
|---|---|---|---|---|---|---|---|---|---|
| | | P ($\hat{E}$ > C) | P($\hat{E}$ = C) | P($\hat{E}$ < C) | Results | P($\hat{E}$ > C) | P($\hat{E}$ = C) | P($\hat{E}$ < C) | Results |
| Gini | 10 | **0.99422** | 0.00578 | 0 | W | 0.11896 | 0.88104 | 0 | – |
| | 50 | **1.00000** | 0 | 0 | W | 0.92820 | 0.07180 | 0 | – |
| | 100 | **0.99842** | 0.00158 | 0 | W | 0.40562 | 0.59438 | 0 | – |
| | 200 | **0.99984** | 0.00016 | 0 | W | 0.61030 | 0.38970 | 0 | – |
| Shannon | 10 | **0.99682** | 0.00318 | 0 | W | 0.36518 | 0.63482 | 0 | – |
| | 50 | **0.99858** | 0.00142 | 0 | W | 0.61088 | 0.38912 | 0 | – |
| | 100 | **0.99630** | 0.00370 | 0 | W | 0.40206 | 0.59794 | 0 | – |
| | 200 | **0.99976** | 0.00024 | 0 | W | 0.48300 | 0.51700 | 0 | – |
| Rényi | 10 | **0.99994** | 0.00006 | 0 | W | 0.54572 | 0.45428 | 0 | – |
| | 50 | **1.00000** | 0 | 0 | W | 0.84586 | 0.15414 | 0 | – |
| | 100 | **0.99994** | 0.00006 | 0 | W | 0.67818 | 0.32182 | 0 | – |
| | 200 | **1.00000** | 0 | 0 | W | 0.62926 | 0.37074 | 0 | – |
| Tsallis | 10 | **0.99326** | 0.00674 | 0 | W | 0.11070 | 0.88930 | 0 | – |
| | 50 | **1.00000** | 0 | 0 | W | 0.87864 | 0.12136 | 0 | – |
| | 100 | **0.99664** | 0.00336 | 0 | W | 0.30874 | 0.69126 | 0 | – |
| | 200 | **0.99964** | 0.00036 | 0 | W | 0.49862 | 0.50138 | 0 | – |

tuple of three posterior probabilities $(\delta_l, \delta_e, \delta_r)$, where each posterior indicates that it is probable that $M_1$ is better than $M_2(\delta_l)$, $M_1$ and $M_2$ are equivalent $(\delta_e)$, or that $M_2$ is better than $M_1(\delta_r)$. In *Benavoli et al. (2017)* it is further defined the term "practically equivalent". In this sense, it is defined by the *region of practical equivalence* (ROPE). For example, a ROPE of 0.01 (assuming a performance metric ranged in $[0, 1]$) represents that two classifiers whose mean difference of performance metric is less than 0.01 (or 1%) are practically equivalent. Moreover, to enable sensible automatic decisions based on the three posterior probabilities returned by the Bayesian signed-rank test, it is suitable to define a threshold ε, such that

- $M_1 \gg M_2$ if $P(M_1 > M_2) > \varepsilon$, $M_1$ is better than $M_2$,
- $M_1 \ll M_2$ if $P(M_1 < M_2) > \varepsilon$, $M_2$ is better than $M_1$,
- $M_1 = M_2$ if $P(M_1 = M_2) > \varepsilon$, $M_1$ and $M_2$ are equivalent,

otherwise, no decision can be made.

The Bayesian signed-rank test was thus applied to the cross-validation results using the baycomp (https://baycomp.readthedocs.io/en/latest/) Python package (baycomp version 1.0.3). Test results are shown in Table 11 for the DT algorithms and in Table 12 for the RF algorithms. We considered ε = 0.95 as in *Benavoli et al. (2017)* and performed tests for a ROPE of 1% and 2%. Moreover, for the proposed method $\hat{E}$ and the classically established entropies C, we consider that

**Table 12 Bayesian signed-rank test for the RF algorithms across cross-validation results of all datasets for the weighted F1-score.** Here, ROPE = {0.01, 0.02}; ε = 0.95; **W** represents a significant win for $\hat{E}$; **D** represents a significant draw for $\hat{E}$; and, **L** represents a significant loss for $\hat{E}$. For the case of significant wins, the posterior probability value is shown in bold.

| C | Depth | Estimators | ROPE = 1% | | | | ROPE = 2% | | | |
|---|---|---|---|---|---|---|---|---|---|---|
| | | | $(\hat{E} > C)$ | $(\hat{E} = C)$ | $(\hat{E} < C)$ | Results | $(\hat{E} > C)$ | $(\hat{E} = C)$ | $(\hat{E} < C)$ | Results |
| Gini | 10 | 10 | **1.00000** | 0 | 0 | W | **0.99812** | 0.00188 | 0 | W |
| | 10 | 25 | **0.99998** | 0.00002 | 0 | W | 0.89364 | 0.10636 | 0 | – |
| | 10 | 50 | **1.00000** | 0 | 0 | W | **0.99958** | 0.00042 | 0 | W |
| | 25 | 10 | **0.99886** | 0.00114 | 0 | W | 0.34598 | 0.65402 | 0 | – |
| | 25 | 25 | **0.99954** | 0.00046 | 0 | W | 0.57040 | 0.42960 | 0 | – |
| | 25 | 50 | **0.99992** | 0.00008 | 0 | W | 0.58474 | 0.41526 | 0 | – |
| Shannon | 10 | 10 | **1.00000** | 0 | 0 | W | **0.99990** | 0.00010 | 0 | W |
| | 10 | 25 | **1.00000** | 0 | 0 | W | **0.95558** | 0.04442 | 0 | W |
| | 10 | 50 | **1.00000** | 0 | 0 | W | **1.00000** | 0 | 0 | W |
| | 25 | 10 | **1.00000** | 0 | 0 | W | **0.97280** | 0.02720 | 0 | W |
| | 25 | 25 | **1.00000** | 0 | 0 | W | 0.93018 | 0.06982 | 0 | – |
| | 25 | 50 | **0.99996** | 0.00004 | 0 | W | 0.84228 | 0.15772 | 0 | – |
| Rényi | 10 | 10 | **1.00000** | 0 | 0 | W | **0.99954** | 0.00046 | 0 | W |
| | 10 | 25 | **1.00000** | 0 | 0 | W | **0.95534** | 0.04466 | 0 | W |
| | 10 | 50 | **1.00000** | 0 | 0 | W | **0.99986** | 0.00014 | 0 | W |
| | 25 | 10 | **1.00000** | 0 | 0 | W | **0.95498** | 0.04502 | 0 | W |
| | 25 | 25 | **0.99968** | 0.00032 | 0 | W | 0.55596 | 0.44404 | 0 | – |
| | 25 | 50 | **0.99886** | 0.00114 | 0 | W | 0.55484 | 0.44516 | 0 | – |
| Tsallis | 10 | 10 | **1.00000** | 0 | 0 | W | **0.99944** | 0.00056 | 0 | W |
| | 10 | 25 | **1.00000** | 0 | 0 | W | **0.99930** | 0.00070 | 0 | W |
| | 10 | 50 | **1.00000** | 0 | 0 | W | **0.99992** | 0.00008 | 0 | W |
| | 25 | 10 | **0.99998** | 0.00002 | 0 | W | 0.81052 | 0.18948 | 0 | – |
| | 25 | 25 | **0.99994** | 0.00006 | 0 | W | 0.42182 | 0.57818 | 0 | – |
| | 25 | 50 | **0.99962** | 0.00038 | 0 | W | 0.81208 | 0.18792 | 0 | – |

- $P(\hat{E} > C) > ε$, is a significant win (**W**) for $\hat{E}$,
- $P(\hat{E}1 = C) > ε$, is a significant draw (**D**) for $\hat{E}$ and $C$,
- $P(\hat{E} < C) > ε$, is a significant loss (**L**) for $\hat{E}$.

# DISCUSSION

Through the analysis of Table 4, it is possible to see that the introduced method shows improvements in comparison with classical entropies across almost all tree structures and datasets. Even with a relatively reduced number of search iterations (100 iterations), we were able to frequently outperform existing methods. These improvements are less noticeable, as expected, in the case of the weaker learner (DTs), where a higher proportion of null LI is found. Nonetheless, it must be considered that metric LI provides a worst-case scenario outcome, where comparisons are always made with the best performance of the existing entropic measures. Generally, the combination of the proposed method with RFs

yields a more substantial increase in performance, with a reduced number of null LI, with the notable exception being the 'wilt' dataset, in which almost all methods showed similarly high performance. Moreover, in terms of HI, it is possible to see improvements of over 40%, especially in the case of the stronger RF models, particularly in the case of the 'Wine Quality'.

In general terms, the highest performance gains seem to occur on multiclass and imbalanced datasets (*e.g.*, 'Wine Quality', 'Statlog'), where the choice of performance metric (weighted F1-score) would be most unfavorable. Another example can be seen in the case of the synthetic datasets, namely 'synth_2' and 'synth_5', which are the two synthetic sets with the highest performance increase, and, simultaneously, the only two multiclass synthetic datasets, albeit with balanced classes. Regarding data types, the performance of the proposed method does not seem to suffer negatively. For instance, in datasets with mixed datatypes (*e.g.*, 'Students'), the introduced criterion is still able to outperform classical methods. The results for the remaining metrics, namely Balanced Accuracy, Precision, and Recall (seen in Tables 5-10) generally show a lesser improvement of the proposed method, compared to the standard methods. This is, however, expected as these metrics do not accurately reflect the large data imbalance that exists in multiple of the tested datasets. Nevertheless, the results provide some additional insights into the method's behavior. Specifically, the gains in weighted F1-score appear to stem not only from a modest overall improvement in classification balance, as reflected in the slight increases in Balanced Accuracy, but also from an improved detection of minority classes, as indicated by the relative improvements in Recall for underrepresented categories.

The largest advantage of the proposed method as a splitting criterion is that it never introduces a performance penalty, and, at worst, it is only capable of equalling the classical measures. This is more clearly shown in Tables 11 and 12. When using the ROPE value of 1% (as suggested in *Benavoli et al. (2017)* and later applied in *Hernández et al. (2021)* for the comparison of split criteria), our method is seen to win against all other methods in the case of DTs and RFs. When increasing ROPE to 2%, we are still able to significantly win half the tests when using the RFs. Note that draws were never significant and that the proposed method never lost against classical ones. Furthermore, the statistical results in Table 12 show that using $\hat{E}$ as a splitting criterion is particularly advantageous on shallow tree ensembles. A possible interpretation for this occurrence is that the low maximum depth, in combination with our criteria $\hat{E}$, makes it so that the classifier can more easily generalize to unseen data.

Although the proposed method suffers from implementation challenges, since the foundational structure for the devised DTs/RFs is not on the level of optimization as seen in established works such as scikit-learn, resulting in considerably higher computing times, it could nonetheless be integrated through the addition of the proposed split criterion component within the algorithms. As such, there had to be a compromise in terms of the number of sampled hyperparameters (both from the introduced parametrization and for conventional DT/RF parameters). Moreover, compared with, for instance, Gini, the computational cost of $\hat{E}$ is much greater due to the introduction of logarithmic functions. This, however, seems like a reasonable middle ground considering the added flexibility

introduced by our method. This can be of particular value when in combination with solutions such as that discussed in *Loyola-González, Ramírez-Sáyago & Medina-Pérez (2023)*, where the dynamic selection of splitting criteria at each split could greatly benefit from a unified parametric expression that could be easily searched through, for instance, a genetic search algorithm.

## CONCLUSIONS

Tree-based classifiers are widely used in practical machine-learning applications. The DT, in particular, despite its simplicity, can have reasonable performance while being easily interpretable. It is also the foundational component for other stronger learning models, such as RFs. These factors still motivate research into the further development of DTs. The work described in this article focuses on the data-splitting process, which is one of the key algorithmic mechanisms of DTs. For this, we introduced a 5-parameter expression that aims to generalize entropies traditionally employed as target functions to partition data at DT nodes. This expression, $\hat{E}$, is capable of, for instance, retrieving the Gini impurity, Shannon entropy, Rényi entropy, and Tsallis entropy. However, because of its flexibility, other entropic estimators can also be extracted from parametric manipulation.

To test its applicability as a splitting criterion, a target function was formulated based on $\hat{E}$ and employed in both DTs and RFs. This criterion was then validated against the most common information-theoretic measures of entropy on multiple datasets (both synthetic and derived from open data repositories), using 5-fold cross-validation and parameter tuning through a reduced 100-iteration Bayesian search. From the preliminary results, two main conclusions can be drawn: (i) performance improvements could reach upwards of 40% in F1-score compared to classical measures of entropy; (ii) in the worst-case scenario, no performance penalization would be introduced through the use of $\hat{E}$ (it would only be equivalent to the best method). To validate the latter of these claims, an additional Bayesian statistical analysis was conducted (signed-rank test), showing that for the conventional ROPE value of 1%, our method only produced significant wins against comparable alternatives. Furthermore, our method never suffered from any significant loss, assessing its assurance to worst-case match performance. These findings underscore the importance of careful selection of a splitting strategy.

As future research paths, multiple potential advancements can be identified, but perhaps the most relevant are: a deeper study into the influence of each parameter in $\hat{E}$ in the overall performance of the classifier, guiding prospective practitioners and reducing computational costs of extensive search spaces; the adaptation of this target function into regression problems; the combination of this method with the dynamic selection of split criteria (as discussed in *Loyola-González, Ramírez-Sáyago & Medina-Pérez (2023)*), taking further advantage of the flexibility introduced by $\hat{E}$; and, the application of the proposed method in streaming data scenarios through the incorporation of $\hat{E}$ into the internal operating structure of Hoeffding Trees.

### Funding

The present study was developed in the scope of the Project "Agenda ILLIANCE" [C644919832-00000035 | Projeto n.°46], financed by PRR–Plano de Recuperação e Resiliência under the Next Generation EU from the European Union. Diogo Costa holds a Postdoctoral fellowship grant, and Vasco Costa holds a PhD grant in the scope of "Agenda ILLIANCE". Diogo Costa and Eugénio M. Rocha were supported by the Center for Research and Development in Mathematics and Applications (CIDMA) through the Portuguese Foundation for Science and Technology FCT-Fundação para a Ciência e a Tecnologia), references UIDB/04106/2020 and UIDP/04106/2020. There was no additional external funding received for this study. The funders had no role in study design, data collection and analysis, decision to publish, or preparation of the manuscript.

### Grant Disclosures

The following grant information was disclosed by the authors:
PRR–Plano de Recuperação e Resiliência: C644919832-00000035.
Portuguese Foundation for Science and Technology (FCT-Fundação para a Ciência e a Tecnologia): UIDB/04106/2020, UIDP/04106/2020.

### Competing Interests

The authors declare that they have no competing interests.

### Author Contributions

- Diogo Costa conceived and designed the experiments, performed the experiments, analyzed the data, performed the computation work, prepared figures and/or tables, authored or reviewed drafts of the article, and approved the final draft.
- Vasco Vieira Costa performed the experiments, analyzed the data, performed the computation work, prepared figures and/or tables, authored or reviewed drafts of the article, and approved the final draft.
- Eugénio Rocha conceived and designed the experiments, analyzed the data, authored or reviewed drafts of the article, and approved the final draft.

### Data Availability

The source code is available at Zenodo: Costa, D. (2025). Tree Algorithms with Custom Information Gain Functions. Zenodo. https://doi.org/10.5281/zenodo.15241909.

The classification datasets are available at Zenodo: Costa, D. (2025). Datasets for Testing Custom Tree Splitting Criteria [Data set]. Zenodo. https://doi.org/10.5281/zenodo.15241934.

### Supplemental Information

Supplemental information for this article can be found online at http://dx.doi.org/10.7717/peerj-cs.3319#supplemental-information.

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
