# Peer review of "A new parametric information-gain criterion for tree-based machine learning algorithms"

_PeerJ Computer Science, doi:10.7717/peerj-cs.3319_

## Round 0.1 · original submission · Major Revisions

· Academic Editor

Major Revisions

The reviewers have substantial concerns about this manuscript. The authors should provide point-to-point responses to address all the concerns and provide a revised manuscript with the revised parts being marked in different color.

Reviewer 1 ·

Basic reporting

The manuscript presents a new method for data partitioning, proposing a criterion that aims to enhance classification performance. While the topic falls within the scope of current research interests in machine learning and data analysis, the contribution of the paper appears to be quite limited.

The novelty of the paper is significantly restricted. The criterion introduced for partitioning includes numerous parameters, and there is no strong theoretical or empirical justification for their inclusion or choice.

Moreover, the motivation behind the new measure is weak and lacks a thorough conceptual foundation. The paper would benefit from a deeper discussion of the theoretical properties of the introduced formula, which is currently missing. This omission weakens the impact and clarity of the proposed contribution.

The criterion for data partitioning - proposed method - is overly complex due to the large number of parameters it contains. The need for an exhaustive grid search, combined with comparisons on a validation set, results in high computational demands. This raises concerns about the practical usability of the method, especially in real-world scenarios with limited computational resources.

Furthermore, the authors fail to analyze or even acknowledge the implications of the criterion’s structure. An exploration of its mathematical properties, potential biases, and behavior under different data distributions would be crucial for a more complete understanding.

Experimental design

The experimental section is incomplete and lacks essential analysis. The evaluation does not consider important performance metrics such as precision, recall (sensitivity), or balanced accuracy, which are crucial for a nuanced assessment of classification quality, especially in imbalanced datasets.

Additionally, the authors do not comment on the observed differences in performance between qualitative, quantitative, and mixed-type datasets. Understanding how the method behaves with different types of data is fundamental and would significantly strengthen the experimental claims.

While the structure of the paper is generally acceptable, the exposition of the core contributions lacks clarity and depth. The introduction could better articulate the motivation behind the proposed method, and the conclusions are not well supported by the presented results.

Validity of the findings

The proposed criterion involves many parameters, which leads to the need for a full grid search. However, the paper does not demonstrate that the search was conducted systematically or that the method generalizes well beyond the specific experimental setup. There is no indication that overfitting to the validation data was avoided, raising concerns about the reliability of the reported performance.

The evaluation focuses on a limited set of performance indicators, omitting crucial ones such as precision, recall (sensitivity), and balanced accuracy. These metrics are essential, particularly when dealing with imbalanced datasets. The absence of such analysis casts doubt on whether the results reflect true improvements or are simply artifacts of specific evaluation choices.

The paper does not examine how the method performs across datasets with different characteristics (qualitative, quantitative, and mixed). Without such analysis, it is unclear whether the method's effectiveness is consistent or heavily dependent on data type.

Additional comments

Overall, while the paper addresses a relevant problem, it does not meet the standards required for publication in its current form. The contribution is limited, the proposed method is computationally expensive and not well motivated, and the experimental evaluation is insufficient and lacks key analyses.

To improve, the authors should:
• Provide a stronger theoretical motivation and analysis of their method.
• Discuss and analyze the computational complexity connected with tuning parameters.
• Evaluate method using a broader set of performance metrics.
• Discuss results across different types of data (qualitative, quantitative, mixed).
• Deepen the discussion of the formula’s properties and the rationale behind the chosen parameters.
• Discuss the properties of the proposed formula.

Reviewer 2 ·

Basic reporting

Congratulations, team, and thank you for sharing this paper. The manuscript is written in clear and professional English with a good logical flow. The terminology used is standard in the machine learning domain, and the paper is accessible to practitioners and researchers with a solid understanding of decision trees and information theory. This paper requires readers with a solid mathematical background to follow the mathematical demonstrations, in particular in the PROOFS OF KNOWN ENTROPIES section.

The manuscript provides thorough and up-to-date references, contextualizing the proposed work effectively within the literature. The citations include foundational works (e.g., Shannon, Gini, Quinlan) and recent advances in parametric entropy and tree-based algorithms.

The figures and tables are generally well-constructed and relevant. Figure 1 is particularly helpful for visualizing the behavior of the proposed entropy under different parameterizations. Tables (e.g., Table 1, Table 4) effectively summarize empirical findings. I think as an improvement, you can ensure that all figures and tables are labeled clearly, consistently referred to in the text, and fully described. Figure 1 could benefit from additional caption details to aid standalone interpretability.

Data and Code links attached in the manuscript are not working and give a 403 error. When we click on them, for example, "The source code is available at Zenodo: Costa, D. (2025). Tree Algorithms with Custom Information
420 Gain Functions. Zenodo. https://doi.org/10.5281/zenodo.15241909
421 The classification datasets are available at Zenodo: Costa, D. (2025). Datasets for Testing Custom
422 Tree Splitting Criteria [Data set]. Zenodo. https://doi.org/10.5281/zenodo.15241934";. Please make sure all these URL links are working.

Experimental design

I think that the study addresses a well-defined and relevant research gap, which is improving decision tree splitting criteria by proposing a new, flexible parametric entropy. This fits well within the broader push for explainable and adaptable ML methods.

The derivation of the new entropy function is mathematically sound and convincingly justified. The authors benchmark their approach against standard criteria (Gini, Shannon, Rényi, Tsallis), using thorough cross-validation and Bayesian statistical testing.

The experiments are performed on a diverse set of datasets (binary, multiclass, balanced, imbalanced), which strengthens the generalizability of the findings.

Areas of Improvement:

- There are a few equations that should be improved:
* Eq 1 should mention that IG is a function of S and A, IG(S,A).
* Eq 2, Shannon entropy is typically defined using base-2 logarithms. This form measures information in bits (binary units), which is standard in information theory, as introduced by Claude Shannon in 1948.

- The methodology is well-documented, including hyperparameter search ranges and cross-validation details. The use of Bayesian optimization for parameter tuning is appropriate and well justified. Suggestion: A schematic or pseudocode of the modified decision tree algorithm could further help readers implement the method without referring to external code.

Validity of the findings

The findings are statistically well-supported. The authors go beyond simple performance comparisons by employing the Bayesian signed-rank test, addressing critiques of NHST approaches. They present both worst-case (LI) and best-case (HI) performance improvements, providing nuanced insights into effectiveness.

The discussion appropriately contextualizes the results. The authors recognize limitations (e.g., computational complexity and unoptimized implementation) and propose future directions.

In terms of improvements:

- I recommend adding a brief sensitivity analysis or visualization of how individual parameters (among the 5 parameters) affect performance or entropy behavior to guide practical hyperparameter tuning.

- The method is evaluated on datasets of varying sizes but does not assess scalability to large-scale data explicitly.

Additional comments

Great paper!!

Reviewer 3 ·

Basic reporting

The article is quite interesting, comparing several datasheets.

However, the abstract is still written in general terms.
Comments are included in the manuscript.

Experimental design

In the introduction, it is necessary to explain the datasheet used and the 40% accuracy result from which it was obtained.

Validity of the findings

-

Additional comments

The abstract needs to be revised.

Annotated reviews are not available for download in order to protect the identity of reviewers who chose to remain anonymous.

---

## Round 0.2 · Major Revisions

· Academic Editor

Major Revisions

There are some remaining concerns that need to be addressed.

Reviewer 1 ·

Basic reporting

no comment

Experimental design

no comment

Validity of the findings

no comment

Additional comments

The authors have addressed my previous comments, particularly those concerning the experimental methodology. The changes introduced have significantly improved the clarity of the validation process description, dataset selection, and parameter optimization approach. My concerns regarding the experimental justification of the proposed splitting criterion's effectiveness have been addressed through an expanded description of the validation procedure and the addition of a statistical analysis based on a Bayesian test.

At the same time, I would like to suggest further refinement of the commentary on the obtained results. Specifically:
- It would be helpful to more clearly indicate for which types of datasets (e.g., balanced vs. imbalanced, binary vs. multiclass, synthetic vs. real-world) the proposed solution offers the greatest benefits. Although examples are mentioned in the text (e.g., Wine Quality, Statlog), a synthetic summary of this observation is missing.
- The paper also presents results for Balanced Accuracy, Precision, and Recall. It would be worthwhile to elaborate on whether these additional metrics lead to any significant conclusions about the method’s behavior, e.g., whether the improvement in F1-score is related to better detection of minority classes or rather to an overall improvement in classification balance.
- I appreciate the authors’ honesty regarding computational limitations. It might be worth considering a brief comment on whether and how the proposed approach could be integrated with existing libraries (e.g., scikit-learn), which would enhance its practical usability.

In summary, the revisions have significantly improved the quality of the paper. I am pleased to note that my previous comments have been taken into account, and the manuscript is now much more complete.

Reviewer 3 ·

Basic reporting

The author has not explained this comment:

Where did this 40% increase come from, and what dataset was used?

It would be best to briefly explain the 40% increase and the dataset used.

Experimental design

no comment

Validity of the findings

no comment

Additional comments

no comment

---

## Round 0.3 · accepted · Accept

· Academic Editor

Accept

Reviewers are satisfied with the revisions, and I concur to recommend accepting this manuscript.

Reviewer 1 ·

Basic reporting

To be honest, the added discussions are rather sparse, do not fully respond to my comments, and fail to provide any significant guidance or analysis for applying the proposed approach. The authors cite space and page limitations as a justification. So I suppose I have no choice but to accept these sparse additions.

Experimental design

No comment

Validity of the findings

No comment

Additional comments

No comment

Reviewer 3 ·

Basic reporting

The author has completed the comments or notes on the manuscript that has undergone the review process.

Experimental design

no comment

Validity of the findings

no comment

Additional comments

no comment